

# Implementation of an IBBCEAS technique in an atmospheric simulation chamber for *in situ* NO₃ monitoring: characterization and validation for kinetic studies

Axel Fouqueau[1], Manuela Cirtog[1], Mathieu Cazaunau[1], Edouard Pangui[1], Pascal Zapf[1], Guillaume Siour[1], Xavier Landsheere[1], Guillaume Méjean[2], Daniele Romanini[2], Bénédicte Picquet-Varrault[1]

[1]LISA, UMR CNRS 7583, Université Paris-Est Créteil, Université de Paris, Institut Pierre Simon Laplace (IPSL), Créteil, France

[2]LIPHY, UMR CNRS 5588, Université Grenoble Alpes, Grenoble, France

*Correspondence to*: Manuela Cirtog (manuela.cirtog@lisa.u-pec.fr)

**Abstract.** An incoherent broadband cavity-enhanced absorption spectroscopy (IBBCEAS) technique has been developed for *in situ* monitoring of NO₃ radicals at the ppt level in the CSA simulation chamber (at LISA). The technique couples an incoherent broadband light source centered at 662 nm with a high finesse optical cavity made of two highly reflecting mirrors. The optical cavity which has an effective length of 82 cm allows for up to 3 km of effective absorption and a high sensitivity for NO₃ detection (up to 6 ppt for an integration time of 10 seconds). This technique also allows NO₂ monitoring (up to 9 ppb for an integration time of 10 seconds). Here, we present the experimental setup as well as tests for its characterization and validation. The validation tests include an intercomparison with another independent technique (FTIR) and the absolute rate determination for the reaction *trans*-2-butene + NO₃ which is already well documented in the literature. The value of $(4.13 \pm 0.45) \times 10^{-13}$ cm³ molecule⁻¹ s⁻¹ has been found, which is in good agreement with previous determinations. From these experiments, optimal operation conditions are proposed. The technique is now fully operational and can be used to determine rate constants for fast reactions involving complex volatile organic compounds (with rate constants up to $10^{-10}$ cm³ molecule⁻¹ s⁻¹).

## 1. Introduction

The night time chemistry in polluted urban or sub-urban areas has been proved to be governed by NO₃ radicals since its discovery in the 1980s *(Naudet et al., 1981; Noxon et al., 1978, 1980; Platt et al., 1980)*. In particular, NO₃ radical has been shown to be an efficient oxidant for some organic compounds, or in some cases even the dominant one, thus impacting the budget of these species and their degradation products. Unsaturated VOCs, including biogenic VOCs, are particularly reactive towards NO₃ radicals *(Wayne et al., 1991)*. Providing kinetic data for these reactions is essential for a better understanding of the role of NO₃ radicals in their degradation. Nevertheless, due to the high reactivity of some unsaturated VOCs with NO₃ (with rate constants which can reach $10^{-11}$ to $10^{-10}$ cm³ molecule⁻¹ s⁻¹), absolute rate determination for these reactions appears to be difficult as it requires the use of a highly sensitive method for NO₃ monitoring. As a consequence, the number of absolute kinetic studies for the NO₃-initiated oxidation of terpenes is very limited and this leads to large uncertainties on this chemistry as it has been pointed out in the literature *(Atkinson, 2000; Brown and Stutz, 2012; Ng et al., 2017)*. *Calvert et al., 2015* gave recommendations for NO₃ oxidation rate constants for 91 alkenes (ranging between $10^{-16}$ and $10^{-10}$ cm³ molecule⁻¹ s⁻¹) and more than 98 % of the determinations on which these recommendations are based were conducted using the relative rate method. One of the reasons to this is still the





challenging measurement of $NO_3$ radicals at low mixing ratios (<100 ppt) during such experiments. For these

compounds, new absolute determinations are essential to better evaluate the role of $NO_3$ radical in their degradation.

Among the various experimental tools which are currently used to measure rate constants, atmospheric simulation chambers represent suitable tools for performing experiments under very realistic atmospheric conditions. This implies low concentrations of reactants in order to minimize possible secondary reactions.

Another benefit of these facilities is the high analytical capabilities which allow *in situ* monitoring of reactants and products with a high sensitivity. Even though significant progresses have been made in the last decades for $NO_3$ radicals measurement at low concentrations with the arising of cavity enhanced and cavity ring down spectroscopic techniques (Ball et al., 2004; Bitter et al., 2005; Kennedy et al., 2011; Langridge et al., 2008) as well as laser induced fluorescence techniques (Matsumoto et al., 2005b, 2005c, 2005a; Wood et al., 2003), it is

observed only few were coupled to simulation chambers (Dorn et al., 2013; Venables et al., 2006; Wu et al., 2014).In addition, to our knowledge, none of these techniques have been used for kinetic applications involving $NO_3$ radical in simulation chambers.

In this purpose, the analytical capabilities of the CSA chamber available at LISA have been improved by developing a sensitive technique for measuring $NO_3$ radicals at very low concentration. An incoherent broadband

cavity enhanced absorption spectroscopy (IBBCEAS) technique has been coupled to the chamber with the objective of performing high sensitivity *in situ* $NO_3$ monitoring with integration time of seconds.

In this paper, we describe the experimental setup and the characterization of the technique. Finally, the IBBCEAS technique has been validated thanks to an intercomparison of $NO_2$ and $NO_3$ measurement with FTIR technique and an absolute rate determination for the well-known reaction *trans*-2-butene+$NO_3$.

**2.    Experimental section**

**2.1. The CSA chamber**

The CSA chamber is made of a large and evacuable Pyrex® reactor (6 m length, 45 cm diameter and 977 L volume) which has been previously presented in details (Doussin et al., 1997). It is equipped with a homogenization system which is made up of an injection pipe (4 meters long, 1 cm diameter and regularly

drilled with 1 mm holes), 2 stainless steel fans and a close-circuit Teflon pump connected at both ends and allows a mixing time below one minute. The chamber is also equipped with two *in situ* spectroscopic analytical devices coupled with "White" type multiple reflection systems inside the reactor: (i) a FTIR spectrometer (Bruker Vertex 80) which allows acquiring spectra in the range of 600-4000 cm$^{-1}$ with a maximal spectral resolution of 0.07 cm$^{-1}$ and an optical path length of 204 m; (ii) an UV-Visible grating spectrometer consisting of

a high-pressure Xenon arc lamp (Osram XBO, 450 W Xe UV), a monochromator HR 320 (Jobin-Yvon) and a CCD camera (CCD 3000, 1024 x 58 pixel, Jobin-Yvon) as a detector. This spectrometer allows acquiring spectra with a spectral resolution of 0.18 nm and an optical path length of 72 m.

This facility has intensively been used to investigate complex gas-phase chemistry involving organic compounds and to provide kinetic and mechanistic data. In particular, it has been used for absolute rate determination of

reactions involving $NO_3$ radicals with a series of VOCs like ethers, esters and aldehydes (Kerdouci et al., 2012; Picquet-Varrault et al., 2009; Scarfogliero et al., 2006). In these studies, $NO_3$ radical was monitored at 662 nm





with the *in situ* UV-visible spectrometer. However, due to the poor detection limit (0.5 ppb for 1 minute of acquisition), and taking into account the experimental conditions, the range of rate constants that can be investigated is limited ($< 10^{-12}$ cm$^3$ molecule$^{-1}$ s$^{-1}$) preventing from studying very reactive chemical systems, such as BVOC+NO$_3$ reactions.

### 2.2. The IBBCEAS setup

In order to improve the analytical capabilities of the CSA chamber, an IBBCEAS has been developed and coupled to the chamber for high sensitivity *in situ* NO$_3$ monitoring. A detailed description of the technique has been provided in previous works (Langridge et al., 2008; Romanini et al., 1997). IBBCEAS measurements are conducted by exciting with an incoherent broad-band source, a high finesse optical cavity formed by two mirrors with high reflectivity (R(λ) ~99.98 %). Photons resonate between the two mirrors increasing their lifetime in the cavity by a factor of 1/(1-R(λ)). During this time, photons traverse an effective path length of kilometers inside the cavity making possible observations of absorbing species at very low concentrations. The intensity transmitted by the optical cavity rapidly reaches steady state. The optical intracavity absorption coefficient of the sample $\alpha(\lambda)$ can then be calculated with the following expression if accurate measurement of cavity reflectivity R(λ) and of the distance between the mirrors (d) is provided (Engel et al., 1998):

$$\alpha(\lambda) = \left(\frac{I_0(\lambda)}{I(\lambda)} - 1\right)\left(\frac{1-R(\lambda)}{d}\right)$$ (Eq. 1)

Here I(λ) and I$_0$(λ) are the transmitted intensities measured in the presence and in absence of the absorbing species respectively.

The concentrations of the absorbing molecules can then be calculated using a least square algorithm to simultaneously fit the molecules absorption cross section using the formula:

$$\alpha(\lambda) = \sum_i [X_i]\, \sigma_{X_i}(\lambda) + p(\lambda)$$ (Eq. 2)

Where, $\sigma_{X_i}(\lambda)$ are the absorption cross sections, [X] are the species absorbing in the considered spectral region and p(λ) is a cubic polynomial to correct baseline deformations due to potential variations of the source intensity (Venables et al., 2006) or to absorption and/or scattering of particles in the chamber (Varma et al., 2013).

The optical cavity is made of two high reflectivity mirrors (Layertec, plan/concave mirrors with a 1 m radius of curvature, nominal reflectivity of 99.98 ± 0.01 % between 630 and 690 nm). It has been transversally installed on the CSA chamber using two co-axial outputs of the reactor. A scheme of the IBBCEAS instrument interfaced to the chamber is shown in **Erreur ! Source du renvoi introuvable.**. The distance between the mirrors is 82 cm and includes 45 cm for the chamber diameter, 2 × 10.5 cm for the Pyrex outputs and 2 × 8 cm for the interface mounts between the chamber and the commercial CRD Optics mount support. In order to prevent from adsorption of semi-volatile species or deposition of particles on the mirrors, which would result in a significant decrease of the reflectivity, the mirrors can be protected thanks to a continuous nitrogen flush (flow rate: 300 mL min$^{-1}$) using a 1/16 inch input disposed close to the mirror surface. By comparing the absorption coefficients measured with and without the flush, for a known quantity of NO$_2$ in the chamber, this effective length was estimated to be d$_{eff}$ = 62 ± 3 cm (i.e. 24% lower than the physical length of the cavity). Thanks to the mixing





system which ensures a fast homogenization of the mixture in the chamber, this effective length was observed to be constant during the whole duration of an experiment.

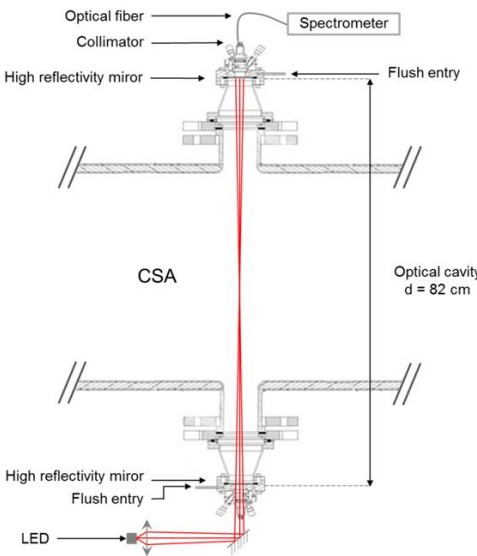

115    **Figure 1 : Scheme (transverse section) of the IBBCEAS instrument interfaced to the CSA chamber**

A light emitting diode (LED, Mouser Electronics – Starboard, Luminus SST-10-DR-B130 DEEp Red, K, D5) with an approximative Gaussian shaped emission of 19 nm full width at half maximum (FWHM) and centered at 662 nm was used in order to monitor $NO_3$ at its maximum absorption wavelength. The LED emission spectrum is compared to the cross section spectra of $NO_3$ (Orphal et al., 2003) and $NO_2$ (Vandaele et al., 1997) in Figure 2.

120    It can be observed that the spectral range of the LED is large enough to allow monitoring both $NO_2$ and $NO_3$.

**Figure 2: (a) $NO_3$ and $NO_2$ absorption cross sections (*Orphal et al., 2003; Vandaele et al., 1997*) between 640 and 680 nm (convolved with apparatus function of the spectrometer) and LED emission spectrum and (b) mirror reflectivity**



The LED is mounted on a thermo electric controller (TEC) device (ThermoElectric Cooling, Laser Mount Arroyo Instruments) to ensure a very precise temperature regulation (± 0.01 °C) and stabilize the spectral distribution of the LED. With this device, changes in the LED intensity have been observed to be below 0.3 %. A laser diode controller (Arroyo Instrument 6310) provides the electric power for both TEC and LED (intensity sent to the LED is fixed at 900 mA). Light emitted by the LED is spatially incoherent and collimation is required for an effective coupling with the optical cavity. The light is hence focused with a convex lens (Thorlabs Aspheric Condenser Lens, 25.4 mm diameter, F = 16 mm, NA = 0.79) and injected into the optical cavity with two concave mirrors (Thorlabs, protected silver, diameter 50.8 mm, f = 50mm and diameter 75 mm, f = 500 mm respectively) in order to focalize the beam in the middle of the cavity. Light transmitted through the cavity is directed thanks to a collimator (Thorlabs SMA Fiber Collimation Pkg, 635 nm, f = 35.41 mm, NA = 0.25) and an optical fiber (Ocean Optics VIS-NIR (200 μm slit, 5m long) to a miniature Ocean Optics QE65000 spectrometer. The spectrometer measures the cavity output wavelength dependent intensity and comprises a spectrograph interfaced to a charged coupled device (CCD) thermally stabilized at -15°C to minimize dark current. The spectral range covered by the spectrometer is 45 nm (640-685 nm) with a spectral resolution of 0.2 nm. In order to calculate the concentrations of absorbing species, a data treatment program has been developed in R (Ihaka and Gentleman, 1996) using a least square algorithm. A third-degree polynomial function is used in the fit to take into account baseline deformation due to small changes in the source intensity. The concentrations of the absorbing species and the polynomial function are fitted by minimizing the RMSE (root mean square error). In practice, the optimization was run following a bound optimization by quadratic approximation (BObyQA) method (Pow-ell, M., 2009). The iterative process to minimize the RMSE (between absorption coefficients from Eq. 1 and Eq.2) stops when none of the parameters vary more than 0.2 % between two successive iterations. In the studied spectral range, absorbing species are $H_2O$, $NO_2$ and $NO_3$. Due to dry conditions used during the experiments, $H_2O$ absorption was considered negligible. The absorption cross sections used are provided by the literature (Orphal et al., 2003; Vandaele et al., 1997) and have been convoluted with the apparatus function of the instrument. Because $NO_2$ cross sections provided by Vandaele et al., 1997 are measured up to 666.5 nm, the treatment has systematically been conducted up to this value. The cross sections used for the data treatment are presented in .

### 3. Technique qualification and characterization

Several experiments have been carried out to assess the stability, the accuracy and the detection limit of the technique. First experiments have been conducted to test the optical stability and the influence of pressure variations on the device. These tests have shown that the instrument is very stable (variations < 1 %). Two aspects have been shown to be particularly critical for measurement with IBBCEAS technique: $I_0$ measurement and the determination of mirror reflectivity.

### 3.1. Determination of the cavity reflectivity

Having a precise knowledge of the wavelength dependent mirror reflectivity, $R(\lambda)$, is one of the most critical point of the IBBCEAS technique (Venables et al., 2006). Two different methods have been proposed for accurate determination of $R(\lambda)$: i) measurement of a known concentration of an absorbing species (Ventrillard-Courtillot et al., 2010), ii) measurement of the ring-down time in the empty cavity using a pulsed laser CRDS





technique (Ball et al., 2004). The first method has been employed here. The absorbing species which has been chosen for the experiments is $NO_2$ as it absorbs in the whole spectral region of the LED emission and its absorption cross sections are known with high accuracy. $NO_2$ concentrations in the simulation chamber were

obtained from *in situ* FTIR measurements using $IBI_{NO2}$ (1530–1680 cm$^{-1}$) = (5.6 ± 0.2) × 10$^{-17}$ cm molecule$^{-1}$ (base e,). To retrieve the mirror reflectivity R($\lambda$) from $NO_2$ concentration, the following equation is used:

$$R(\lambda) = 1 - d \times \sigma_{NO_2}(\lambda) \times [NO_2] \times \left(\frac{I_0(\lambda)}{I(\lambda)} - 1\right)^{-1} \qquad \text{(Eq. 3)}$$

where $\sigma_{NO_2}(\lambda)$ is the $NO_2$ absorption cross section (Vandaele et al., 1997) and $[NO_2]$ is the concentration of $NO_2$ determined by FTIR. In order to reduce the uncertainty on the reflectivity determination and to compensate the

weak cross sections of $NO_2$ in the 660–670 nm region, high concentrations (up to 800 ppb) were used. A plot showing the variation of the reflectivity in function of the wavelength is presented as an example in . Due to the $NO_2$ reference spectrum, the reflectivity is measured up to 666.5 nm. During this experiment, the reflectivity was observed to vary between 99.975 at 640 nm and 99.974 % at 667 nm and this is in agreement with the reflectivity provided by the supplier (99.98 ± 0.01 %). It was found to have a slight dependence with

wavelengths ($y = -4.5 \times 10^{-7}x + 1.000039$), which justifies its measurement on a wide wavelengths range. At 662 nm, which corresponds to the maximum of $NO_3$ absorption, the reflectivity was found to be 99.974 %. At this wavelength, the effective absorption path length estimated from Eq. 4 is found to be 3.4 km:

$$X(\lambda) = d/\left(1 - R(\lambda)\right) \qquad \text{(Eq. 4)}$$

In addition, it has been observed that the reflectivity of the cavity can significantly vary with the mirror

cleanliness. As an example, successive experiments showed that reflectivity can vary at 662 nm from 99.974 % to 99.971 % from one experiment to another, leading to variations of the effective absorption path length of almost 20 %. Therefore, it is crucial to precisely determine the reflectivity prior of each experiment.

**3.2. $I_0(\lambda)$ measurement**

Previous studies (Fuchs et al., 2010; Kennedy et al., 2011; Ventrillard-Courtillot et al., 2010) have pointed out

that the $I_0(\lambda)$ has to be periodically recorded during an experiment to ensure accurate measurement with the IBBCEAS technique. Indeed, changes in the lamp emission spectrum or poor optical stability may induce changes in the absorption coefficient and therefore generate errors in the quantification of the species. This fact may be an issue for experiments in simulation chambers as the $I_0(\lambda)$ can only be recorded before injecting the reactants and experiments can then last for several hours.

In order to evaluate the stability of the signal during a typical experiment and the uncertainty generated by the use of a unique $I_0(\lambda)$ on the quantification of the species, experiments have been performed by injecting $NO_2$ into the chamber and by monitoring its concentration with the IBBCEAS technique for several hours. After the chamber has been filled with a mixture of $N_2/O_2$ (80/20) at atmospheric pressure, the $I_0(\lambda)$ has been measured. Then, $NO_2$ was introduced into the chamber with mixing ratios ranging between 100 ppb and 1 ppm and the

signal I($\lambda$) was measured. From these measurements, the absorption coefficient and the $NO_2$ concentration have been calculated and plotted as a function of time in . An increase up to 3 % on $NO_2$ concentration has been observed 2 hours after recording the $I_0(\lambda)$ due to the deviation of the baseline which is no longer well corrected by the polynomial function. These results suggest that the accuracy of the measurement is significantly reduced



after 2 hours. The length of the experiments should therefore not exceed this duration. Above this limit the
uncertainty due to a unique measurement of the $I_0(\lambda)$ can be considered as negligible.

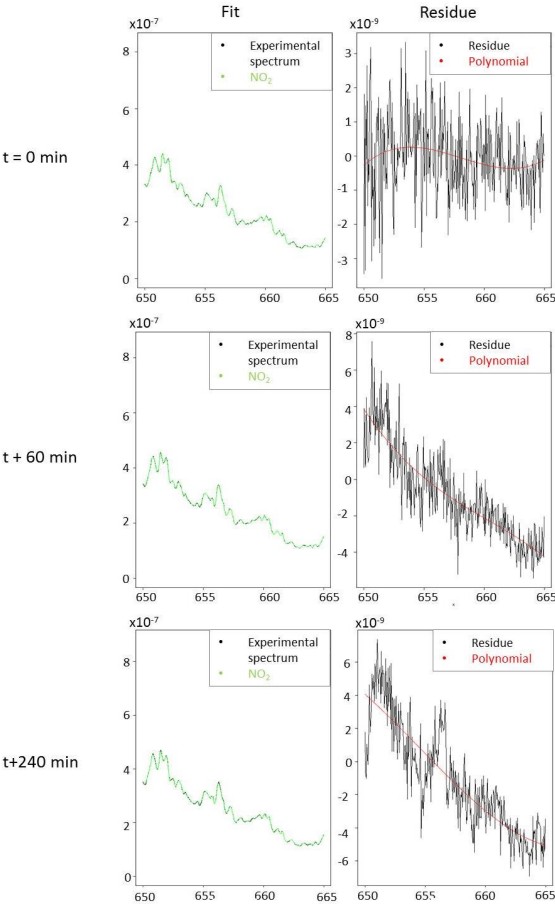

**Figure 3 : Evolution of fit quality as a function of time since $I_0$ was made.**

### 3.3. Detection limit and Allan variance

The detection limit for $NO_3$ radical was estimated by considering 3 times the peak-to-peak noise on the
absorption coefficient at 662 nm, which corresponds to the maximum absorption of $NO_3$ radical. For 10 seconds
of acquisition time (which corresponds to 25 acquisitions of 400 ms), it has been found to be $1.2 \times 10^{-9}$ cm$^{-1}$.
Considering that $NO_3$ radical cross section at this wavelength is $2.2 \times 10^{-17}$ cm$^2$ molecule$^{-1}$, the detection limit for
$NO_3$ was estimated to 6 ppt. The same approach was used to estimate the detection limit for $NO_2$. Between 645
and 650 nm, which correspond to the two main absorption peaks of $NO_2$, noise has been found to be $2 \times 10^{-9}$ cm$^{-1}$
. Considering the difference of maximum and minimum absorption of $2.2 \times 10^{-20}$ cm$^2$ molecule$^{-1}$ in this range,
detection limit has been found to be 11 ppb for 10s of integration time. A spectrum measured with an acquisition



time of 10 s, for 6 ppt of $NO_3$ and 630 ppb of $NO_2$ is shown for illustration in . The used fit range was between

655 and 666.5 nm. Even at this low level of $NO_3$ concentration, the absorption is clear and allows its quantification. The fit appears to be satisfying for both $NO_2$ and $NO_3$ and the residual spectrum appears to be essentially made of noise, showing a good efficiency of the polynomial fit and a satisfying subtraction of species contributions. This figure shows that this wavelength range is efficient for a precise detection and quantification of both species and validates the detection limit.

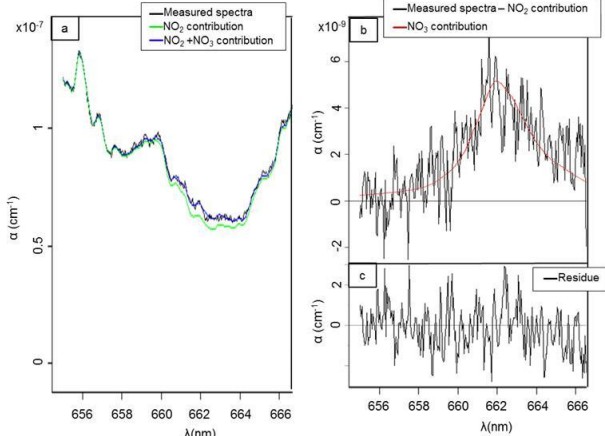

**Figure 4: (a) measured absorption coefficient α(λ) (between 655 and 669 nm) with integration time of 10 s (in black); complete fit of $NO_2$ and $NO_3$ (in blue) with [$NO_3$] = 6 ppt; [$NO_2$] = 630 ppb; $NO_2$ fit only (in green) ; (b) measured absorption coefficient (in black) without $NO_2$ contribution and fitted with [$NO_3$] = 6 ppt (in red); (c) residue of measured and fitted absorption coefficient.**

The potential of the IBBCEAS technique for measuring $NO_3$ radical during simulation chamber experiments has

already been explored in previous works. It has been coupled to the simulation chamber at UCC (Cork, Ireland), to SAPHIR chamber at FZJ, (Jülich, Germany) and to CHARME chamber at LPCA (Wimereux, France). The characteristics and performances of the various instruments are compared in . Our instrument exhibits very good performances with detection limit similar to those of the other instruments, but for shorter integration time and/or for smaller effective length. This reflects the very good stability of the optical system. These results also

prove the potential of this technique for measuring $NO_3$ radical at low level of concentrations with a good time resolution (10 s) and thus suitable for kinetic measurements.

**Table 1 : Comparison of characteristics and performances of various IBBCEAS coupled to simulation chambers for the detection of $NO_3$ radicals.**

| *In situ* IBBCEAS | Deff* (cm) | DL / Integration time | Reference |
|---|---|---|---|
| **LISA, Créteil France** | **82** | **6 ppt / 10 s** | **Current work** |
| UCC, Cork, Ireland | 462 | 4 ppt / 57 s | Venables et al., 2006 |
| UCC, Jülich, Germany** | 1800 ± 20 | 0.5-2 ppt/5 s | (Dorn et al., 2013) |
| LPCA, Wimereux, France | 2000 | 7.9 ppt / 1 min | Wu et al., 2014 |





*Deff is the effective length of the cavity, calculated by taking into account the dilution generated by the mirror protective flush

**UCC's IBBCEAS was used on SAPHIR chamber during an intercomparison campaign.

In order to evaluate if our detection limit can be improved by increasing the integration time, the Allan variance has been calculated for various integration times during an experiment in which $NO_2$ concentration was monitored. The $NO_2$ mixing ratio was approximatively 1300 ppb. The Allan variance $\sigma_A^2$ is given by the equation:

$$\sigma_A^2 = \frac{1}{2(M-1)} \sum_{i=1}^{M-1} [x_{i+1}(t_{av}) - x_i(t_{av})]^2 \qquad \text{(Eq.5)}$$

with M the number of measurements, $t_{av}$ the integration time and x the concentration of $NO_2$ measured. In this experiment, 30 000 measurements of 2 seconds have been performed and the Allan variance was then calculated for various integration times ranging between 2 and 4096 seconds. The standard deviation of Allan, defined as the square root of the Allan variation provides an indication of the instrument stability in time. It is plotted as a function of the integration time in . For very short integration time (few seconds) the Allan deviation is very high due to the white noise of the instrument. The Allan deviation decreases with increasing integration time until 100 seconds. For longer integration times, the deviation increases with increasing integration time. Nevertheless, the deviation is low, showing that the instrument is very stable. Due to this stability, we expect that the detection limits calculated before can be improved by increasing integration time. From this test, it can be also concluded that the optimal integration time is around 100 seconds.

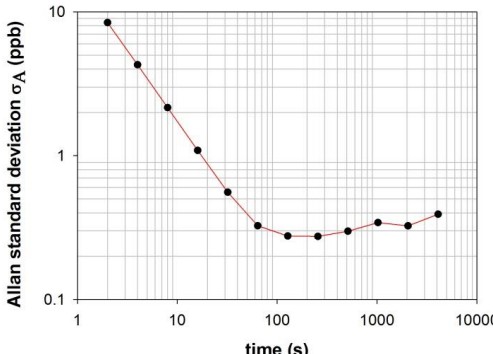

**Figure 5: Allan variance vs. integration time calculated for the IBBCEAS technique**

The high stability of the Allan deviation after 100 s also suggests that the stability of the optical device is optimal to perform measurements without recording a new $I_0(\lambda)$ for at least 4096 s, in agreement with the result of the test presented in section 3.2. In conclusion, the good stability of the optical device complies with the constraint of experiments in simulation chamber where $I_0(\lambda)$ can only be recorded at the beginning of the experiment.

### 3.4. Determination of the uncertainty

Considering Eq.1 and Eq. 2, the overall relative error on $NO_3$ concentration can be considered as the square root of the sum of the square relative errors, the reflectivity $R(\lambda)$ and the $NO_3$ absorption cross sections. In case of the

use of the nitrogen flush, the uncertainty on $d_{eff}$ has to be also taken into account (8 %, see Sect. 2). Considering Eq. 3, the uncertainty on R(λ) should include the uncertainty on $NO_2$ concentration measured by FTIR estimated to be 8 % as well as the uncertainty on $NO_2$ absorption cross sections. The uncertainties on $NO_2$ and $NO_3$ cross

sections are estimated to be 3 % in the spectral range of interest (Vandaele et al. 1997 and Orphal, Fellows and Flaud, 2003). However, the uncertainty generated by the data treatment Δfit, i.e. by the fit, which results mainly from the noise in the spectra, should also be taken into account. Because the nitrate radical is an unstable species, this uncertainty cannot be estimated by calculating the standard deviation on its concentrations measured for a long period of monitoring. It was therefore estimated by considering the noise of a $NO_3$ concentration profile and

has been found to be ~ 3ppt for 10 seconds of integration time. The overall absolute error on $NO_3$ concentration is then expressed by the following formula:

$$\Delta N_{NO_3} = \sqrt{\left(\left(\frac{\Delta\sigma_{NO_2}(\lambda)}{\sigma_{NO_2}(\lambda)}\right)^2 + \left(\frac{\Delta N_{NO_2,refl}}{N_{NO_2,refl}}\right)^2 + \left(\frac{\Delta\sigma_{NO_3}}{\sigma_{NO_3}}\right)^2\right)} \times N_{NO_3} + \Delta fit \qquad (Eq. 6)$$

where $\frac{\Delta\sigma_{NO_2}(\lambda)}{\sigma_{NO_2}(\lambda)}, \frac{\Delta N_{NO_2,refl}}{N_{NO_2,refl}}, \frac{\Delta\sigma_{NO_3}}{\sigma_{NO_3}}$ are relative uncertainties on $NO_2$ cross sections, $NO_2$ concentration used for the reflectivity measurement, $NO_3$ cross sections respectively and $N_{NO_3}$ the concentration of $NO_3$. For 10 seconds of

integration time, the uncertainty is thus 9 % with an absolute part of 3 ppt.

**4. Intercomparison study**

After having defined the optimal operation conditions of the IBBCEAS, the technique has been validated thanks to an intercomparison with another instrument. During a dedicated experiment, $NO_3$ and $NO_2$ concentrations were measured by the IBBCEAS technique while $NO_2$ and $N_2O_5$ were monitored by *in situ* FTIR. The chamber

was first filled dry synthetic air ($N_2/O_2 - 80/20$) at atmospheric pressure and $I_0(\lambda)$ spectra were recorded for both FTIR and IBBCEAS. Several hundred of ppb of $NO_2$ (Air Liquide N20, purity>99 %, $H_2O$<3000 ppm) were then introduced into the chamber and IBBCEAS spectra were recorded in order to determine the mirrors reflectivity R(λ) (see Sect. 3.2).

Then, by considering the following equilibrium:

$NO_2 + NO_3 + M \rightarrow N_2O_5 + M\ (k_1)$ (R. 1)

$N_2O_5 + M \rightarrow NO_2 + NO_3 + M\ (k_2)$ (R. 2)

And by assuming that this equilibrium is reached, $NO_3$ concentration can be deduced from $NO_2$ and $N_2O_5$ ones measured by FTIR. This hypothesis appears justified as no other reactive species has been introduced into the chamber and may thus disturb the equilibrium. The equilibrium constant ($K = k_1/k_2$) is well known and has

been shown to highly depend on temperature and pressure (Atkinson et al., 2004). These parameters were therefore precisely monitored during the experiment leading to the value of $2.17 \times 10^{-11}$ $cm^3.molecule^{-1}$ at 298K and at 1030 mbar. IBBCEAS spectra were recorded every 30 seconds while FTIR ones were recorded every 2 minutes. The integrated band intensities used to quantify $NO_2$ and $N_2O_5$ with FTIR were: $IBI_{NO2}$ (1530-1680 $cm^{-1}$) = $(5.6 \pm 0.2) \times 10^{-17}$ cm molecule$^{-1}$ and $IBI_{N2O5}$ (1200-1285 $cm^{-1}$) = $(4.05 \pm 0.4) \times 10^{-17}$ cm molecule$^{-1}$ (base e).

The correlation plots between FTIR and IBBCEAS for $NO_3$ and $NO_2$ measurements are shown in . $NO_2$



concentrations measured by the two techniques are in very good agreement (the maximum difference between the two techniques is 6 %), with a slope of 1.0. The intercept of the linear regression (b = 15.0 ppb) is not significantly different from zero as it is lower than its uncertainty calculated as twice the standard deviation (Δb = 25.1 ppb). For FTIR measurements, uncertainties on $NO_2$ were calculated as the sum of relative uncertainties

on IBI and on the spectra treatment (10 %). For IBBCEAS, the error was estimated as the square root of the sum of the square uncertainties on the reflectivity and on the $NO_2$ cross sections (9 %). For $NO_3$ radical, the concentrations obtained by the two techniques are also in good agreement for the whole range of concentrations, from few ppt to several hundred ppt. The slope of the linear regression is 1.1 suggesting a bias of 10 % between the two techniques. However, this difference is not significant in regards to the uncertainties which are

represented by the black dashed lines. For FTIR measurement, the uncertainties are calculated being the error on $NO_2$ and $N_2O_5$ measurement and on the equilibrium constant (21 %). The calculation for uncertainty on $NO_3$ IBBCEAS measurement is presented in section 3.4. The intercept appears to be very low (around 3 ppt).

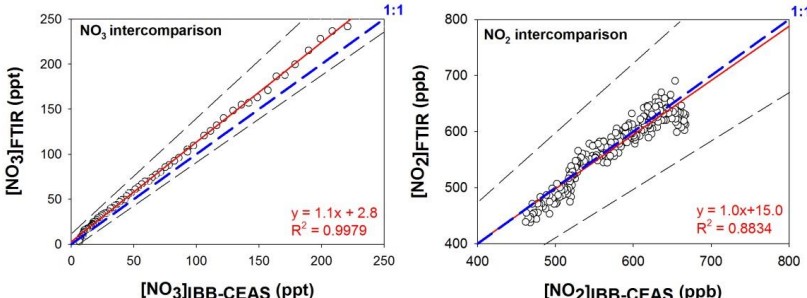

**Figure 6 : Correlation between FTIR and IBBCEAS measurements for $NO_3$ (left graph) and $NO_2$ (right graph).**
**Uncertainties are shown by dashed straight lines. Blue dashed lines show the 1:1 ratio.**

In conclusion, the IBBCEAS exhibits very good agreement with the FTIR, for both $NO_3$ and $NO_2$ monitoring, with good sensitivity. This agreement is very satisfactory considering the fact that the IBBCEAS samples across the reactor width while the FTIR provides an integrated measurement on the whole reactor length.

**5.   Kinetic study: $NO_3$ + *trans*-2-butene**

The last step of the validation consisted in a kinetic experiment in order to assess the potential of the technique for kinetic studies: the IBBCEAS has been used to measure the rate constant of a well-known reaction: *trans*-2-butene + $NO_3$. This reaction has been chosen as it has been intensively studied in the literature. Six absolute rate determinations (Benter et al., 1992; Berndt et al., 1998; Dlugokencky and Howard, 1989; Kasyutich et al., 2002; Ravishankara and Mauldin, 1985; Rudich et al., 1996) and two relative ones (Atkinson et al., 1984; Japar and

Niki, 1975) have been published leading to a recommendation by IUPAC (Atkinson et al., 2006). This will allow us to test the performances of the instrument for monitoring $NO_3$ concentrations with a high time resolution and to validate our kinetic determination by comparison with previous ones.

The rate constant was determined using the absolute rate technique and by measuring the consumption of *trans*-2-butene due to its reaction with $NO_3$. Because no other oxidant was present in the mixture, it was therefore

assumed that *trans*-2-butene is consumed only by reaction with nitrate radical:



$$trans\text{-}2\text{-}butene + NO_3 \rightarrow Products \ (k_{BVOC}) \tag{R. 3}$$

For this reaction, kinetic equation can be established as:

$$\frac{-d[trans\text{-}2\text{-}butene]}{dt} = k_{BVOC}[NO_3][trans\text{-}2\text{-}butene] \tag{Eq. 7}$$

By making the hypothesis of small variations of time and [*trans*-2-butene], this relationship can be approximated

to:

$$-\Delta[trans\text{-}2\text{-}butene] = k_{BVOC}[NO_3][trans\text{-}2\text{-}butene]\Delta t \tag{Eq. 8}$$

Where $\Delta[trans\text{-}2\text{-}butene]$ corresponds to the consumption of *trans*-2-butene during the time interval $\Delta t$ and [*trans*-2-butene] and [$NO_3$] are averaged concentrations during this period. By plotting $-\Delta$[*trans*-2-butene] vs. the product [*trans*-2-butene][$NO_3$]$\Delta$t, a straight line with the slope corresponding to $k_{BVOC}$ is obtained.

Six kinetic experiments have been conducted in the dark, at room temperature (292 - 294 K) and atmospheric pressure in synthetic air. The initial conditions of reactants (*trans*-2-butene, $N_2O_5$, $NO_2$) are listed in . The reflectivity was measured prior to each experiment by introducing $NO_2$ into the chamber. When present, $NO_2$ initial concentrations were used also to slow down the reaction by shifting the $N_2O_5$ equilibrium. *Trans*-2-butene (Air Liquide, purity>99 %) was then introduced and it was checked that no significant loss was observed in the

timescale of the experiment. Nitrate radicals were generated into the simulation chamber from the thermal decomposition of dinitrogen pentoxide. $N_2O_5$ was stepwise injected in order to assure a consumption of *trans*-2-butene in a proper timescale to satisfactory monitor the reactants. Time resolved concentrations of *trans*-2-butene, $NO_2$ and $N_2O_5$ were monitored from their infrared absorption spectra every 2 minutes. The integrated band intensity used to quantify the VOC is IBI$_{trans\text{-}2\text{-}butene}$ (870-1100 cm$^{-1}$) = (2.8 ± 0.3) × 10$^{-18}$ cm molecule$^{-1}$

(base e, measured by previous internal work). $NO_3$ was monitored with the IBBCEAS technique with an acquisition time of 30 seconds.

**Table 2 : Injected mixing ratios for the kinetic study of the reaction *trans*-2-butene + NO₃**

| Experiment | [NO$_2$]$_0$ (ppb) | [N$_2$O$_5$] (ppb) × number of injections | [*trans*-2-butene]$_0$ (ppb) |
|---|---|---|---|
| 1 | / | 2500 | 1920 |
| 2 | / | 300 × 2 ; 150 × 2 | 750 |
| 3 | 920 | 1000 × 3 | 990 |
| 4 | 950 | 1500 × 2 | 1110 |
| 5 | 750 | 300 × 3 | 1110 |
| 6 | / | 300 × 3 | 1030 |

shows time profiles of reactants during a typical experiment. At the moment when $N_2O_5$ is injected, a rapid

decrease of *trans*-2-butene and $NO_3$ concentrations is observed together with a large production of $NO_2$ due to $N_2O_5$ decomposition.





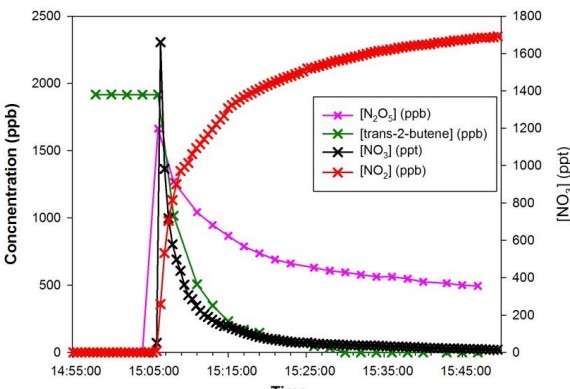

**Figure 7 : Concentrations of *trans*-2-butene and NO$_2$ measured by FTIR (left axis) and NO$_3$ measured by IBBCEAS versus time (right axis) during experiment 1.**

The kinetic plot (−Δ[*trans*-2-butene] vs. the product [*trans*-2-butene]×[NO$_3$]×Δt) gathering data from all experiments is presented in . The uncertainty on each experimental point was calculated as the sum of the relative uncertainties on [*trans*-2-butene] and [NO$_3$] for the abscissa scale (the uncertainty on the time was considered to be negligible) and as twice the uncertainty on the [*trans*-2-butene] for the ordinate scale. From this figure, it can be observed that all experiments are in good agreement. In consequence, a linear regression was

performed on all data points leading to a rate constant of $(4.13 \pm 0.45) \times 10^{-13}$ cm$^3$ molecule$^{-1}$ s$^{-1}$. The uncertainty on the rate constant was estimated as twice the standard deviation on the linear regression. The obtained rate constant has been compared to the values from previous determinations and to the value recommended by IUPAC in Table 3.

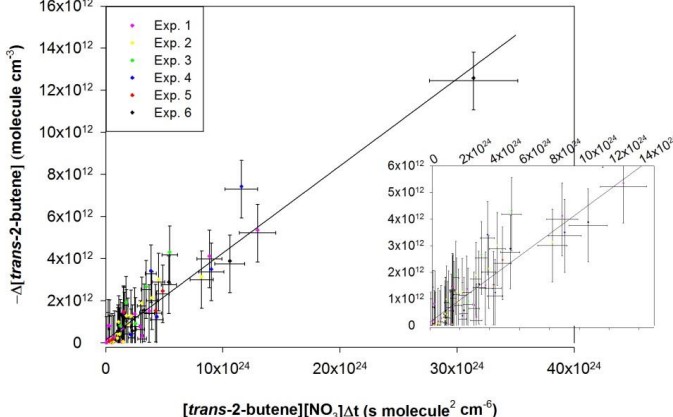

**Figure 8 : Absolute kinetic plot for the reaction of *trans*-2-butene with NO$_3$ radical, showing the loss of trans-2-butene vs. [VOC]×[NO$_3$]×Δt. To the right, a zoom on the points close to the origin.**



This new determination appears to be in very good agreement with the IUPAC recommendation and with previous absolute determinations, within the uncertainty. However this work does not agree with the two relative determinations which are up to 45 % lower than our determination. Nevertheless, these two values are in

disagreement with all of the previous absolute determinations. These relative rate determinations are relative to the equilibrium constant $K$ ($N_2O_5$ + $M$ ⇆ $NO_2$ + $NO_3$). A possible explanation to this disagreement would be that $NO_3$ concentration is overestimated because it was considered that the equilibrium is reached. However, the reaction with *trans*-2-butene is fast enough to significantly disturb the equilibrium and prevent it to be established. An overestimation of $NO_3$ concentrations would hence lead to an underestimation of the rate

constant.

**Table 3 : Comparison of the rate constant obtained for the reaction of *trans*-2-butene with $NO_3$ with previous determinations.**

| k (cm³ molecule⁻¹ s⁻¹) | T (K) | Technique* | Reference |
|---|---|---|---|
| $(4.13 \pm 0.45) \times 10^{-13}$ | 293 | ($N_2O_5$/CEAS) | This study |
| $(3.90 \pm 0.78) \times 10^{-13}$ ($\Delta \log k = \pm 0.08$) | 298 | recommendation | IUPAC |
| $(3.78 \pm 0.17) \times 10^{-13}$ | 298 | (AR/CEAS) | (Kasyutich et al., 2002) |
| $(3.74 \pm 0.45) \times 10^{-13}$ | 298 | (AR/LIF) | (Berndt et al., 1998) |
| $(4.06 \pm 0.36) \times 10^{-13}$ | 298 | (AR/LIF) | (Rudich et al., 1996) |
| $(3.88 \pm 0.30) \times 10^{-13}$ | 298 | (AR/MS) | (Benter et al., 1992) |
| $(3.96 \pm 0.48) \times 10^{-13}$ | 298 | (AR/LIF) | (Dlugokencky and Howard, 1989) |
| $(3.78 \pm 0.17) \times 10^{-13}$ | 298 | (AR/LIF) | (Ravishankara and Mauldin, 1985) |
| $(3.09 \pm 0.27) \times 10^{-13}$ | 298 | (RR**) | (Atkinson et al., 1984) |
| $(2.31 \pm 0.17) \times 10^{-13}$ | 300 | (RR**) | (Japar and Niki, 1975) |

*Indicates kinetic method (AR = absolute rate, RR = relative rate) and $NO_3$ measurement technique: CEAS, LIF (Laser-Induced Fluorescence) or MS (mass spectrometry) used

** Relative rate determinations are relative to the equilibrium constant $K$ ($N_2O_5$ + $M$ ⇆ $NO_2$ + $NO_3$ + $M$).

In conclusion, this agreement shows that the determination made with the IBBCEAS technique presented in this paper is correct, allowing reliable measurement of $NO_3$ at low concentration with good sensitivity and time resolution. This technique is now operational for application to other absolute kinetic studies.

**6.   Conclusions**

An IBBCEAS technique has been developed and coupled to the CSA simulation chamber for the *in situ* measurement of $NO_3$ radicals at the ppt level. This instrument allows also monitoring $NO_2$ in the ppb range. Thanks to various tests, the instrument has been carefully characterized in order to identify potential bias and to define the optimal operation conditions. The performances of the instrument in terms of detection limit and

uncertainties were also determined. The instrument exhibits very good detection limit for $NO_3$ radicals (6 ppt) for 10 seconds of integration time. This detection limit fully complies with our needs for kinetic applications.

The instrument was also validated thanks to an intercomparison experiment with the *in situ* FTIR technique. With this technique, $NO_3$ concentration was indirectly obtained by monitoring $NO_2$ and $N_2O_5$ concentrations and by using the well-known equilibrium constant $K$ ($N_2O_5$ + $M$ ⇆ $NO_2$ + $NO_3$). The concentrations measured by



the two techniques were shown to be in very good agreement (better than 10%) for both $NO_3$ and $NO_2$, over a ride range of concentrations: from ppt to ppt range for $NO_3$ radical and from ppb to hundreds of ppb for $NO_2$.

Finally, this technique was used for the absolute rate determination of a well-documented reaction, *trans*-2-butene+ $NO_3$. The value of $(4.13 \pm 0.45) \times 10^{-13}$ cm$^3$ molecule$^{-1}$ s$^{-1}$ found in this study is in very good agreement with the previous absolute determinations. Moreover, the good sensitivity and the good time resolution represent

excellent performances allowing the use of this technique for monitoring the $NO_3$ radicals when involved in fast reactions.

The IBBCEAS technique is now operational and will be used in further works, particularly to monitor $NO_3$ concentrations for absolute rate determinations of fast reactions of volatile organic compounds with $NO_3$ radicals.

*Data availability.* Rate constant for the $NO_3$ oxidation of *trans*-2-butene is available Table 3. It is also available through the Library of Advanced Data Products (LADP) of the EUROCHAMP data center (https://data.eurochamp.org/data-access/ gas phase-rate-constants/, last access: 25 March 2020, Fouqueau et al., 2020a). Simulation chamber experiments which were used to retrieve these parameters and for the intercomparaison are available through the Database of Atmospheric Simulation Chamber Studies (DASCS) of

the EUROCHAMP data center (https://data.eurochamp.org/data-access/chamber-experiments/, last access: 25 March 2020, Fouqueau et al., 2020b).

*Author contributions.* MCi, BPV and AF designed the IBBCEAS technique with the help of GM and DR. AF installed and did characterization experiments with the technical support of XL (optical development), MCa and

EP (experiments), PZ and GS (data treatment). MCi, BPV and AF wrote the article and AF was responsible for the final version. All coauthors revised the content of the original manuscript and approved the final version of the paper.

*Competing interests.* The authors declare that they have no conflict of interest.


*Financial support.* This work was supported by the French national programme LEFE/INSU and by the European Commission's Seventh Framework Programme (EUROCHAMP-2; grant no. 228335), H2020 Research Infrastructures (EUROCHAMP-2020; grant no. 730997).

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
