# Peer review of "Implementation of an IBBCEAS technique in an atmospheric simulation chamber for *in situ* NO3 monitoring: characterization and validation for kinetic studies"

_Atmospheric Measurement Techniques, 2020_

## Referee Comment (RC1) · Anonymous Referee #1 · 21 May 2020

Fouqueau et al., AMTD: Implementation of an IBBCEAS technique in an atmospheric simulation chamber for in situ NO3 monitoring: characterization and validation for ki-netic studies

Fourqueau et al. present the development and characterisation of an IBBCEAS system and its application to NO3 kinetic studies of VOCs in an atmospheric simulation cham-ber. NO3 was one of the earliest atmospheric targets of optical cavities using ringdown and broadband approaches in the visible region of the spectrum. This work is not new in that respect, nor in the application of such instruments to atmospheric simulation

chambers. Nevertheless, their work goes further than others in applying this method to the determination of absolute rate coefficients in chamber-kinetic studies of VOC oxidation by NO3. The reported rate coefficient agrees well with other absolute rate coefficients reported in the literature.

While the work is probably eventually publishable, there are omissions and questions about calculations that first need to addressed. There are also quite a large number of grammatical errors or typos that should be corrected.

1. There are multiple missing references to figures and tables:

L.104, L.150, L.171, L.196, L.213, L.227, L.247, L.295, L.336, L.349, L.356

2. Further technical details should be supplied about

L.116: LED optical power

L.146: specific RH range of the experiments

L.280-90: To form NO3, either O3 is needed with NO2 to form NO3, or N2O5 must be added as a precursor. This information is missing in the description. The authors should supply these experimental details along with relevant chemical equations.

L.270: The NO3 concentration should be stated explicitly. Presumably the concentration was zero? How would other absorbing species like NO2 influence the fit variability?

3. It is not clear how the authors calculated certain values. These figures need to be checked:

L.177: I calculate 3.15 km for R = 0.99974 and d = 0.82 m, whereas the authors report 3.4 km. Also, it should also be clarified that this is the pathlength across the full optical cavity length (82 cm) and includes the purge volumes. When purging, the effective pathlength would be about 25% shorter.

L.182: I calculate 12% variation, not 20%

4. The authors should comment briefly on photolysis of NO3 by the IBBCEAS probe beam, potential surface losses of reactants, and the influence of water vapour and aerosols on the spectral analysis. These issues are likely unimportant for kinetic experiments, but should be noted for completeness.

5. It is not entirely clear how the effect of I0 stability has been determined. Is the 3% change in NO2 concentration seen at all concentrations, or is it a typical or worst case scenario? Can losses to walls influence the analysis? The residual in Fig. 3 looks very much like the NO2 absorption spectrum, so why does fitting with the NO2 cross-section not eliminate these features better?

6. For the detection limit calculation, the authors should specify whether this calculation is for a purged or unpurged system.

7. The calculation of the detection limit is based on the signal-to-noise ratio for a single wavelength. However, the spectral fit is constrained by many independent signal measurements at different wavelengths. The authors should evaluate how the multiplex nature of the measurement affects the system's detection limit.

8. Figure 6: At low pptv, it looks like there's still an IBBCEAS signal for NO3 but no FTIR signal. Does this divergence reflect a difference in instrument sensitivities, or equilibrium/heterogeneous chemistry, or some other cause? Please comment.

9. L.313: Are spatial inhomogeneities in gas concentrations expected in the chamber?

10. Figure 7. NO3 and butene losses are very rapid. Would data quality be increased by selecting reaction conditions to produce a slower reaction and more data points?

11. Most of the uncertainty in the kinetics data arises from the uncertainty in the FTIR results. Calibrated CIMS or PTR-MS might be better suited for co-measurement of VOCs. The authors may want to discuss these considerations for their kinetic studies.

Minor corrections:

[Figure]

L.27: either "the NO3 radical has. . ." or "NO3 has. . ."

L.38: "one of the reasons for this. . ."

L.46: "progress has been made"

L.53: "For this purpose,"

L.63: "presented in detail"

L.76: "NO3 was"

L.101: "planar/concave"

L.106: "prevent adsorption of. . ."

L.115: Caption Caption should note that collimating lenses and curved mirrors are not shown on the beam injection side.

L.127: "LED current is fixed at 900 mA"

L.132: "to focus the beam"

L.143: hyphen in reference

L.175: "justifies...wide scale". It is unclear what is meant by this. That it should be measured over a wide wavelength range, or that it is not necessary to do so?

L.182: "prior to each experiment"

L.196: "up to 3 % in NO2 concentration"

Figure 3. Change "residue" to "residual" in figure titles and legends.

L.280: "was first filled with dry synthetic air"

L.281: "Air Liquide NO2,"

L.287: NO2 and N2O5 concentrations

L.300: in IBI and in the spectral treatment

L389: performance

L.396: Clarify "from ppt to ppt range for NO3"

L.400: Monitoring NO3 radicals

L.409: intercomparison

---

## Referee Comment (RC2) · Anonymous Referee #2 · 2 Jun 2020

The paper reports on the implementation of an IBBCEAS technique for the detection of NO3. A few questions arise

1. The kinetics analysis ignores any loss of NO3 other than bimolecular reaction of NO3 with Alkene. This is most probably an over simplification. For example, what about wall loss, there could also be diffusion outside of the analysis region. Using a simple model, the addition of a first order loss of NO3, still results in a straight line from a plot of the change in alkene vs the product of [alkene].[NO3] delta t (as in figure 8), however the retrieved rate coefficient is lowered and this impact gets worse as the wall

loss increases. So if there is wall loss the retrieved rate coefficient does not equate to the bimolecular rate coefficient. How have the authors taken into account wall loss or any other first order loss process? Have they modeled their system? Can they show what the quantitative impact on the retrieved rate coefficient will be? Have they made any attempt to evaluate first order losses of NO3?

2. The authors claim that there is good agreement with literature, however, their rate coefficient is the fastest that has been reported in the literature. Also, given that this rate coefficent is probably not simply the rate coefficient for the bimolecular process, most probably a lower limit as result of other first order loss proves, the true bimolecular rate coefficient in their system is likely to be even faster. Faster rate coefficients normally worry kineticists, if one looks at the absolute rates reported in table 3 the unweighted average is 3.87 × 10-11 with a standard deviation of 0.12 × 10-11 i.e. statistically the reported rate coefficient is higher than the current experimental database. Can the authors explain why this is the case?

3. In Table 1 Deff is given as 82 which is simple the geometric distance, if I am not mistaken. Will that be the case? A purge flow is used to protect the mirrors and this will impact the effective path length, this is also likely to be a function of pressure and flow of purge gas. Have experiments been performed to quantify the Deff as a function of purge gas flow rate and pressure?

4. On line 290 the authors state "these parameters were therefore precisely monitored during the experiment leading to the value of 2.17 × 10-11 cm3.molecule-1 at 298K and at 1030 mbar." Is that the value of the equilibrium constant? If so, what are the errors? They need to be stated. Can the authors show that within experimental error that agrees with those reported in Atkinson et al., 2004, indeed how does it compare to the recommended IUPAC /JPL recommended values?

5. In Figure 6, the authors report linear regression between the FTIR and the BBCEAS . Can they provide errors on the slopes? Also include a description of those errors, e.g.

are they 1 sigma just from the linear fit, or do they take into other experimental errors?

6. Finally what are the absolute errors on the rate coefficient? The authors report a simple error analysis based on the line of best fit, i.e. what is the total error?. The authors need to take into account errors in flows, absorption cross section etc etc

---

## Author Comment (AC1) · 12 Jul 2020

First of all, the authors would like to thank the anonymous referee for this discussion and its constructive comments, corrections and suggestions that ensued. We have carefully replied to all its comments and the paper has been improved following its recommendations. All technical corrections suggested by the referee have been carefully performed. Answers have also been provided for all comments and changes have been performed accordingly. Please find below the answers to the comments:

[Figure]

1. There are multiple missing references to figures and tables: L.104, L.150, L.171, L.196, L.213, L.227, L.247, L.295, L.336, L.349, L.356

The authors thank the anonymous referee for having pointed out these mistakes, which are probably due to an error during the submission process. All the references have been added as suggested (L.104, L.152, L.173, L.202, L.220, L. 234, L.257, L.311, L. 367, L.380, L.387).

2. Further technical details should be supplied about L.116: LED optical power

It was indeed not clearly given, but the letter K in the description of the LED indicates the bin of radiant flux (K being between 390 and 430 mW). We modified the text as follows to add this information: P.2.2 L.117: "max. 430 mW".

L.146: specific RH range of the experiments

Information on relative humidity has been provided P2.2 L.148: "(RH < 1 %)" and P.4 L.289: "The chamber was first filled with dry synthetic air (RH < 1%)".

L.280-90: To form NO3, either O3 is needed with NO2 to form NO3, or N2O5 must be added as a precursor. This information is missing in the description. The authors should supply these experimental details along with relevant chemical equations.

As suggested by the referee, NO3 generation protocol is now explained in the manuscript (The following text has been added between P.4 L.293 and L.298): "NO3 radicals were then formed in situ, using thermal dissociation of N2O5 (R. (1)), which was synthesized in a vacuum line following the reaction between O3 and NO2 (R. (2) and R. (3)). This protocol was adapted from (Atkinson et al., 1984; Schott and Davidson, 1958) and is detailed in Picquet-Varrault et al., 2009.(reactions)"

L.270: The NO3 concentration should be stated explicitly. Presumably the concentration was zero? How would other absorbing species like NO2 influence the fit variability?

It was probably not clear enough in the manuscript, but this value was estimated by

considering the noise of a time profile of NO3 concentration. The concentration was thus not zero. A precision has been added in the text (P.3.4 L.279): "NO3 concentration time profile". To answer the second question, it should be said that, in this wavelength range, spectral signatures of NO2 and NO3 are very different. Furthermore, NO2 is well constrained by several thin absorption bands. The fit program handles the subtraction of NO2 very well and the fit of NO3 is thus not strongly impacted by its absorption. The fit variability due to NO2 influence has been verified by constraining on the same spectrum different values of NO2 concentration ($\pm$ 100 ppb of the real value). The uncertainty on NO2 fit can lead to an uncertainty of max. 5 ppt on NO3 fit. This value appears to be very satisfying, considering that an error of 100 ppb on NO2 concentrations is very unlikely.

3. It is not clear how the authors calculated certain values. These figures need to be checked: L.177: I calculate 3.15 km for R = 0.99974 and d = 0.82 m, whereas the authors report 3.4 km. Also, it should also be clarified that this is the pathlength across the full optical cavity length (82 cm) and includes the purge volumes. When purging, the effective pathlength would be about 25% shorter.

The effective path length was calculated using Eq. 4. The value of 3.4 km is indeed an error, and it has been corrected in the manuscript (P.3.4 L.185).

This value was calculated for a full optical cavity length (82 cm) and does not include the purge volumes. A sentence has been added to specify that the purge was not used during this study (P.2.2 L.111): "Nitrogen flush was not used in this study, but is available for further type of experiments." The effective pathlength when purging is indicated in P.2.2 L.110.

L.182: I calculate 12% variation, not 20%

It was a mistake and it has been corrected in the manuscript (P.3.1 L.183).

4. The authors should comment briefly on photolysis of NO3 by the IBBCEAS probe

beam, potential surface losses of reactants, and the influence of water vapor and aerosols on the spectral analysis. These issues are likely unimportant for kinetic experiments, but should be noted for completeness.

In the used wavelength range, only one photolysis reaction can occur: for $\lambda < 710$ nm, NO3 is dissociated into NO and O2. Nevertheless, it has been shown that for $\lambda > 640$ nm, quantum yield of photolysis is close to 0, and thus suspected close to 0 at 662 nm. But even though NO3 was slightly subject to photolysis locally, the volume enlightened by the probe beam is very small in comparison to the overall volume of the chamber (< 0.04%). The homogenization system allowed a mixing with the rest of the volume. Finally, the intercomparison experiment shows that this eventual loss is not significant because it would have led to lower concentrations measured by the IBBCEAS than the FTIR. So the kinetic method is not impacted by a loss of NO3 due to photolysis. A precision has been added P.4 L.330: "Finally, the intercomparison experiment shows that an eventual loss due to photolysis of NO3 by the beam is not significant because it would have led to lower concentrations measured by the IBBCEAS than by the FTIR. In addition, for wavelength longer than 640 nm, which is the case here, Johnston et al., 1996 have shown that photolysis quantum yield is close to 0. It is thus expected that the photolysis of NO3 in the used wavelength range is not occuring. Furthermore even though NO3 was slightly subject to photolysis locally, the volume enlightened by the probe beam is very small in comparison to the overall volume of the chamber (< 0.04%) and the homogenization system allowed a mixing with the rest of the volume." The authors expect that wall losses of NO3 are occurring. Nevertheless, as mentioned by the referee, the absolute kinetic method consists of measuring the decay of trans-2-butene and to use the NO3 concentration measured during the decay. Then, NO3 additional losses do not affect the rate constant determination. A sentence was added in the manuscript in order to precise this point (P.5 L.360): "It is important to notice that the absolute kinetic method used consists of measuring the decay of trans-2-butene for a known concentration of NO3, and not the decay of NO3 radicals for a known concentration of the VOC. The method is thus not affected by NO3 additional loss processes

(e.g. wall losses, reactions with NO2 or with peroxy radicals). Only additional losses of the VOC would lead to an overestimation of the rate constant. This was checked prior to the experiments (i.e. before in the injection of N2O5) and no significant loss of the VOC was observed in the timescale of the experiment (see below)." There is a very strong water vapor absorption in the wavelength range used. Nevertheless, all experiments have been performed in dry conditions and no water features were present on the spectra. A sentence has been had P.2.2 L.147:"Absorption by water vapor may be very high under atmospheric conditions." Aerosols can also affect the IBBCEAS technique at two levels: i) in general, deposition of aerosols on the mirrors lead to a decrease of their reflectivity. That has been dealt with the nitrogen purge in case of the generation of aerosols. ii) The absorption and scattering of the beam by aerosols can lead to a major decrease of the signal in the used wavelength range. In this study, no SOA was formed in the chamber, causing no such problem. In case of the use of the technique with a system which produces SOA, experimental conditions have been modified by using the nitrogen purge close to the mirrors surface and by reducing the production of SOA (by reducing the concentration of precursor).

5. It is not entirely clear how the effect of I0 stability has been determined. Is the 3% change in NO2 concentration seen at all concentrations, or is it a typical or worst case scenario? Can losses to walls influence the analysis? The residual in Fig. 3 looks very much like the NO2 absorption spectrum, so why does fitting with the NO2 cross-section not eliminate these features better?

The impact of I0 stability has been evaluated by performing 2 types of experiments: First, the stability of the optical system has been verified. For this purpose, long term measurements of the signal have been done, leading to intensity variations lower than 0.3% and to very small baseline distortions. Then, the impact of these baseline distortions on the quantification of the absorbing species has been evaluated by measuring the evolution in time of the concentration of a stable species, here NO2. A mixture of 1 ppm of NO2 in synthetic air was used. A sentence has been added in the manuscript

(P.3.2 L.193): "two types of experiments have been performed: first, the stability of the optical system has been verified. For this purpose, long term measurements of the signal have been performed, leading to variations lower than 0.3% and to very small baseline variations. Second, to verify the impact of these variations on quantification of the absorbing species, experiments were conducted [. . .] Then, a concentration of NO2 was introduced into the chamber (mixing ratios ranging between 100 ppb and 1 ppm, depending of the experiment) and the signal I(ïĄň) was measured" and P.3.2 L.202: "for a concentration of 1 ppm of NO2." It looks very unlikely that wall loss can influence the analysis, because the concentration is measured and decay would have been observed and NO2 is very stable in the CSA chamber. In addition, NO2 concentration is not constrained for the fit so in the case NO2 concentration was not stable, the fitted concentration would be lower. The referee is right when saying that the residual looks like NO2 absorption spectra. It can be explained by a deformation of the baseline in time, due to the remoteness of I0, which prevents the software to fit correctly the NO2 spectra. It is likely that the residual is composed of both this baseline deformation and the small part of NO2 cross section that is not correctly subtracted. The uncertainty generated by this phenomenon is said in the manuscript P.3.2 L.202 (up to 3 % for NO2 concentration measurements).

6. For the detection limit calculation, the authors should specify whether this calculation is for a purged or unpurged system.

The detection limit was calculated without the purge system. It has been specified P.3.3 L.214 in the manuscript: "and an unpurged system,"

7. The calculation of the detection limit is based on the signal-to-noise ratio for a single wavelength. However, the spectral fit is constrained by many independent signal measurements at different wavelengths. The authors should evaluate how the multiplex nature of the measurement affects the system's detection limit.

We indeed used the classical method to determine the detection limit: it has been

calculated on the signal-to-noise ratio for a range of wavelength between 655 nm and 666 nm, and we used the max. value of NO3 absorption. This spectral range presents high absorption of NO3 and weak absorption of NO2, as said in the answer to the question 2, L.270. It should be said that in real conditions, as shown in figure 4 in the manuscript, 6 ppt of NO3 are very close to the detection limit. The determination of the detection limit is thus coherent. Furthermore, this figure shows a spectrum with 630 ppb of NO2, which is an important concentration. The detection limit is not degraded by the absorption of NO2.

8. Figure 6: At low pptv, it looks like there's still an IBBCEAS signal for NO3 but no FTIR signal. Does this divergence reflect a difference in instrument sensitivities, or equilibrium/heterogeneous chemistry, or some other cause? Please comment.

By looking at the data, it seems that this case is for the first two points ([NO3]FTIR = 3.0 ppt; [NO3]IBBCEAS = 6.5 ppt and [NO3]FTIR = 4.6 ppt; [NO3]IBBCEAS = 6.7 ppt). For higher concentrations, the agreement is very good. It seems reasonable to think that the sensitivity of the FTIR measurement can explain this difference: it is based on the measurement of NO2 and N2O5. At this moment, N2O5 concentrations are small (approx. 8 ppb) which is close to the detection limit of the FTIR for this species (approx. 6 ppb). The uncertainty of this measurement is thus high, which can explain this difference at very small concentration of NO3. Please note also that these concentrations are close to the detection limit of the IBBCEAS, so it will not be used in the frame of a kinetic experiment.

9. L.313: Are spatial inhomogeneities in gas concentrations expected in the chamber?

During experiments, a homogenization system is used to prevent from inhomogeneities of the mixtures. It is constituted in three parts: (i) an injection pipe, allowing injecting all along the chamber, ii) two fans, allowing a homogenization of gas inside the chamber and iii) a close-circuit homogenization pump, which samples in one extremity to inject the mixing back in the other one. Experiments have been conducted with an

inert gas (NO2) in order to measure the mixing time. Measurements were done in several points of the chamber. This system thus allows a mixing time inferior to a minute. The good agreement between the two measurement methods in the intercomparison experiment, which measure in two different areas, shows also that there are no major inhomogeneities. A sentence has been had to explain it better in the manuscript P.2.1 L.61: "It is equipped with a homogenization system which is made of i) an injection pipe (4 meters long, 1 cm diameter and regularly drilled with 1 mm holes) which allows to inject the reactants all along the chamber, ii) 2 stainless steel fans allowing a homogenization of gas inside the chamber and iii) a close-circuit Teflon pump connected at both ends allowing a recirculation of the gas mixing. This system allows a mixing time below one minute."

10. Figure 7. NO3 and butene losses are very rapid. Would data quality be increased by selecting reaction conditions to produce a slower reaction and more data points?

For the kinetic of trans-2-butene, it was not necessary to increase the frequency of the measurement because the rate constant is relatively low. For faster kinetics, experimental conditions can be changed, in particular by reducing VOC concentration or increasing the measurement frequency.

11. Most of the uncertainty in the kinetic data arises from the uncertainty in the FTIR results. Calibrated CIMS or PTR-MS might be better suited for co-measurement of VOCs. The authors may want to discuss these considerations for their kinetic studies.

We fully agree with the referee that PTR-MS measurements might have been more precise. Unfortunately, this instrument was not available in our group when we performed the experiments.

Minor corrections:

L.27: either "the NO3 radical has. . ." or "NO3 has. . ."

It has been done.

L.38: "one of the reasons for this. . ."

It has been done.

L.46: "progress has been made"

It has been done.

L.53: "For this purpose,"

It has been done.

L.63: "presented in detail"

It has been done.

L.76: "NO3 was"

It has been done.

L.101: "planar/concave"

It has been done.

L.106: "prevent adsorption of. . ."

It has been done.

L.115: Caption Caption should note that collimating lenses and curved mirrors are not shown on the beam injection side.

It has been done.

L.127: "LED current is fixed at 900 mA"

It has been done.

L.132: "to focus the beam"

It has been done.

L.143: hyphen in reference

It has been done.

L.175: "justifies...wide scale". It is unclear what is meant by this. That it should be measured over a wide wavelength range, or that it is not necessary to do so?

The sentence has been changed to: "which justifies that it is necessary to measure it on a wide wavelengths range".

L.182: "prior to each experiment"

It has been done.

L.196: "up to 3 % in NO2 concentration"

It has been done.

Figure 3. Change "residue" to "residual" in figure titles and legends.

It has been done.

L.280: "was first filled with dry synthetic air"

It has been done.

L.281: "Air Liquide NO2,"

N20 (should be read N-20) is actually the name of the cylinder.

L.287: NO2 and N2O5 concentrations

It has been done.

L.300: in IBI and in the spectral treatment

It has been done.

L389: performance

It has been done.

L.396: Clarify "from ppt to ppt range for NO3"

It has been done.

L.400: Monitoring NO3 radicals

It has been done.

L.409: intercomparison

It has been done.

---

## Author Comment (AC2) · 12 Jul 2020

First of all, the authors would like to thank the anonymous referee for this discussion and its constructive comments, corrections and suggestions that ensued. We have carefully replied to all its comments and the paper has been improved following its recommendations. All technical corrections suggested by the referee have been carefully performed. Answers have also been provided for all comments and changes have been performed accordingly. Please find below the answers to the comments:

[Figure]

1. The kinetics analysis ignores any loss of NO3 other than bimolecular reaction of NO3 with Alkene. This is most probably an over simplification. For example, what about wall loss, there could also be diffusion outside of the analysis region. Using a simple model, the addition of a first order loss of NO3, still results in a straight line from a plot of the change in alkene vs the product of [alkene].[NO3] delta t (as in figure 8), however the retrieved rate coefficient is lowered and this impact gets worse as the wall loss increases. So if there is wall loss the retrieved rate coefficient does not equate to the bimolecular rate coefficient. How have the authors taken into account wall loss or any other first order loss process? Have they modeled their system? Can they show what the quantitative impact on the retrieved rate coefficient will be? Have they made any attempt to evaluate first order losses of NO3?

We may have not explained clearly enough the kinetic method, because it appears that the referee did not understand it. NO3 is indeed subject to losses other than the reaction with the VOC, for example wall losses in the chamber, reaction with NO2, with peroxy radicals. . . Nevertheless, this method consists of monitoring the decay of the trans-2-butene and not the one of NO3, and this decay is plotted as a function of [VOC]x[NO3]xdt. Here, [NO3] is measured with the IBBCEAS technique as a function of the time (and not modelled), so we don't care about additional loss processes. What is important here is the measured concentration of NO3 radicals by the time the consumption of the VOC is measured. So, NO3 can be consumed by other processes without affecting the result. Bu if the VOC is consumed by other loss processes (e.g. wall losses), this could affect the results. For this reason, trans-2-butene stability has been checked before every oxidation, and no significant loss beside dilution was observed. It allowed the authors making the hypothesis that trans-2-butene was only consumed by reaction with NO3.

2. The authors claim that there is good agreement with literature, however, their rate coefficient is the fastest that has been reported in the literature. Also, given that this rate coefficient is probably not simply the rate coefficient for the bimolecular process, most

probably a lower limit as result of other first order loss proves, the true bimolecular rate coefficient in their system is likely to be even faster. Faster rate coefficients normally worry kineticists, if one looks at the absolute rates reported in table 3 the unweighted average is 3.87 × 10-11 with a standard deviation of 0.12 × 10-11 i.e. statistically the reported rate coefficient is higher than the current experimental database. Can the authors explain why this is the case?

We are a bit surprised by this comment. Isn't (3.87 ± 0.12) in agreement with (4.13 ± 0.78)? Agreement between values has to be considered by taking into account the uncertainties, otherwise it has no sense. And by taking them into account, our determination is in agreement with the unweighted average of absolute determinations and with the IUPAC recommended value. But it is also in agreement with all individual absolute determinations. The only disagreement which has been noticed is with relative determinations which are significantly lower than other determinations and where not taken into account for the IUPAC recommended value. In conclusion the authors affirm that our determination is in good agreement with literature.

3. In Table 1, Deff is given as 82 which is simple the geometric distance, if I am not mistaken. Will that be the case? A purge flow is used to protect the mirrors and this will impact the effective path length, this is also likely to be a function of pressure and flow of purge gas. Have experiments been performed to quantify the Deff as a function of purge gas flow rate and pressure?

All experiments presented here were performed without a purge flow so the effective distance was considered to be the geometric distance (82 cm). A sentence clarifying this statement has been added in the manuscript (P.2.2 L.111): "Nitrogen flush was not used in this study, but is available for further type of experiments. " Nevertheless, experiments have been conducted to quantify the impact of purge gas flow and pressure:

- Due to nitrogen purge, pressure is only slightly rising: the purge flow is 300 mL min-1, and 300 mL represents a variation of 0.03 % of the total volume of the chamber.

[Figure]

This flow is compensated by the sampling of measurement instruments. No pressure variation was ever measured due to the purge gas flow. Nevertheless, characterization experiments using a known NO2 concentration were conducted, in order to quantify the impact of major pressure variation on the pathlength. They showed no significant impacts of pressure on the pathlentgth. Pressure variations of 10 mbar in the chamber lead indeed to decrease of the path length of less than 1 %.

- Using also a known concentration of NO2, experiments have been performed by adding different flows of purge, in order to determine an efficient flow, i.e. which does not induces a major decrease of the pathlength and a progressive dilution in the measurement area. They showed that the optimal flow is 300 mL min-1, because it efficiently protects the mirrors and induce no long term dilution. With this flow, Deff is decreasing to 62 cm (as it is written in the manuscript L.110), adding an uncertainty of 5 % to the nitrate radical concentration. A sentence to explain these experiments has been added in the manuscript P.2.2 L.108: "This flow rate has been optimized in order to efficiently protect the mirrors while limiting the dilution of the mixture in the measurement area."

4. On line 290 the authors state "these parameters were therefore precisely monitored during the experiment leading to the value of $2.17 \times 10$-11 cm3.molecule-1 at 298K and at 1030 mbar." Is that the value of the equilibrium constant? If so, what are the errors? They need to be stated. Can the authors show that within experimental error that agrees with those reported in Atkinson et al., 2004, indeed how does it compare to the recommended IUPAC /JPL recommended values?

The value of $2.17 \times 10$-11 cm3.molecule-1 is indeed the equilibrium constant. This value was not measured in our study, but calculated using the parametrization recommended by the IUPAC (Atkinson et al., 2004) at a pressure and temperature which were measured during the experiments. So, here, it appears that there is a misunderstanding. By "parameters", we mean temperature and pressure and not the rate constant. In order to make it clearer, the sentence has been modified P.4 L. 305 in the

manuscript: "These two parameters were therefore precisely monitored during the experiment allowing calculating an equilibrium constant of 2.17 × 10-11 cm3.molecule-1 at 298K and at 1030 mbar, using IUPAC database parameters (Atkinson et al., 2004)."

5. In Figure 6, the authors report linear regression between the FTIR and the BBCEAS. Can they provide errors on the slopes? Also include a description of those errors, e.g. are they 1 sigma just from the linear fit, or do they take into other experimental errors?

Errors on the slopes are now provided. The values were calculated to be 1.0 ± 0.2 for NO2 and 1.1 ± 0.3 for NO3 by considering the statistical error on the slope, which is twice the standard deviation, and the sum of the systematic relative errors on FTIR and IBBCEAS measurements. To explain it, sentences have been written in P.4 L.313: "Here, the overall uncertainty was calculated as the sum of the statistical error on the slope (twice the standard deviation, 4 %) and systematic errors on FTIR (i.e. on IB-INO2, 4 %) and IBBCEAS measurements (which includes uncertainties on NO2 cross sections and on the mirrors reflectivity, 9%)."and in P.4 L.320 "The error is calculated with the same method as for NO2."

6. Finally what are the absolute errors on the rate coefficient? The authors report a simple error analysis based on the line of best fit, i.e. what is the total error? The authors need to take into account errors in flows, absorption cross section etc etc

The authors have considered that the statistical error, calculated as twice the standard deviation on the linear regression and which takes into account the dispersion of the experimental points, is mainly due to spectra treatment uncertainties. However, we agree that this statistical error does not include systematic ones, such as errors on cross sections. So we have added the uncertainty on NO3 concentration. As described in the manuscript, this uncertainty was estimated to be 9% and corresponds to the sum of NO2 and NO3 cross sections errors and the uncertainty of NO2 concentration for reflectivity measurement. By summing the statistical error (10%) and the uncertainty on NO3 concentration (9%), we obtain an overall uncertainty of 0.78 ïĆť 10-13 cm3
molecule-1 s-1. This has been corrected in the manuscript L.391 and in Table 3. It must be noticed that because trans-2-butene concentration is both on x and y axes, the error on IBI is not considered here. An explanation has been added P.5 L.392: "The uncertainty on the rate constant was estimated as the sum of the relative uncertainties on NO3 concentrations and twice the standard deviation on the linear regression."
* * *

---

## Author Response (AR1)

[revised manuscript text omitted]

**Answer to Anonymous Referee #1**

First of all, the authors would like to thank the anonymous referee for this discussion and its constructive comments, corrections and suggestions that ensued. We have carefully replied to all its comments and the paper has been improved following its recommendations. All technical corrections suggested by the referee have been carefully performed. Answers have also been provided for all comments and changes have been performed accordingly. Please find below the answers to the comments:

**1. There are multiple missing references to figures and tables: L.104, L.150, L.171, L.196, L.213, L.227, L.247, L.295, L.336, L.349, L.356**

The authors thank the anonymous referee for having pointed out these mistakes, which are probably due to an error during the submission process. All the references have been added as suggested (L.103, L.152, L.173, L.202, L.220, L. 234, L.257, L.311, L. 367, L.380, L.387).

**2. Further technical details should be supplied about**

**L.116: LED optical power**

It was indeed not clearly given, but the letter K in the description of the LED indicates the bin of radiant flux (K being between 390 and 430 mW).

We modified the text as follows to add this information: P.2.2 L.117: **"max. 430 mW**".

**L.146: specific RH range of the experiments**

Information on relative humidity has been provided P2.2 L.148: "**(RH < 1 %)**" and P.4 L.289: "**The chamber was first filled with dry synthetic air (RH < 1%)**".

**L.280-90: To form $NO_3$, either $O_3$ is needed with $NO_2$ to form $NO_3$, or $N_2O_5$ must be added as a precursor. This information is missing in the description. The authors should supply these experimental details along with relevant chemical equations.**

As suggested by the referee, $NO_3$ generation protocol is now explained in the manuscript (The following text has been added between P.4 L.293 and L.298): "**$NO_3$ radicals were then formed *in situ*, using thermal dissociation of $N_2O_5$ (R. (1)), which was synthesized in a vacuum line following the reaction between $O_3$ and $NO_2$ (R. (2) and R. (3)). This protocol was adapted from (Atkinson et al., 1984; Schott and Davidson, 1958) and is detailed in Picquet-Varrault et al., 2009.**

$$N_2O_5 + M \leftrightharpoons NO_3 + NO_2 + M \tag{R. 1}$$
$$O_3 + NO_2 \rightarrow NO_3 + O_2 \tag{R. 2}$$
$$NO_3 + NO_2 + M \rightarrow N_2O_5 + M \tag{R. 3}$$ "

**L.270: The $NO_3$ concentration should be stated explicitly. Presumably the concentration was zero? How would other absorbing species like $NO_2$ influence the fit variability?**

It was probably not clear enough in the manuscript, but this value was estimated by considering the noise of a time profile of $NO_3$ concentration. The concentration was thus not zero. A precision has been added in the text (P.3.4 L.279): "**$NO_3$ concentration time profile**".

To answer the second question, it should be said that, in this wavelength range, spectral signatures of $NO_2$ and $NO_3$ are very different. Furthermore, $NO_2$ is well constrained by several thin absorption bands. The fit program handles the subtraction of $NO_2$ very well and the fit of $NO_3$ is thus not strongly impacted by its absorption. The fit variability due to $NO_2$ influence has been verified by constraining on the same spectrum different values of $NO_2$ concentration ($\pm$ 100 ppb of the real value). The uncertainty on $NO_2$ fit can lead to an uncertainty of max. 5 ppt on $NO_3$ fit. This value appears to be very satisfying, considering that an error of 100 ppb on $NO_2$ concentrations is very unlikely.

**3. It is not clear how the authors calculated certain values. These figures need to be checked:**

**L.177: I calculate 3.15 km for R = 0.99974 and d = 0.82 m, whereas the authors report 3.4 km. Also, it should also be clarified that this is the pathlength across the full optical cavity length (82 cm) and includes the purge volumes. When purging, the effective pathlength would be about 25% shorter.**

The effective path length was calculated using Eq. 4. The value of 3.4 km is indeed an error, and it has been corrected in the manuscript (P.3.4 L.179).

This value was calculated for a full optical cavity length (82 cm) and does not include the purge volumes. A sentence has been added to specify that the purge was not used during this study (P.2.2 L.111): "**Nitrogen flush was not used in this study, but is available for further type of experiments.**"

The effective pathlength when purging is indicated in P.2.2 L.110.

**L.182: I calculate 12% variation, not 20%**

It was a mistake and it has been corrected in the manuscript (P.3.1 L.183).

**4. The authors should comment briefly on photolysis of $NO_3$ by the IBBCEAS probe beam, potential surface losses of reactants, and the influence of water vapor and aerosols on the spectral analysis. These issues are likely unimportant for kinetic experiments, but should be noted for completeness.**

In the used wavelength range, only one photolysis reaction can occur: for $\lambda < 710$ nm, $NO_3$ is dissociated into NO and $O_2$. Nevertheless, it has been shown that for $\lambda > 640$ nm, quantum yield of photolysis is close to 0, and thus suspected close to 0 at 662 nm. But even though $NO_3$ was slightly subject to photolysis locally, the volume enlightened by the probe beam is very small in comparison to the overall volume of the chamber (< 0.04%). The homogenization system allowed a mixing with the rest of the volume. Finally, the intercomparison experiment shows that this eventual loss is not significant because it would have led to lower concentrations measured by the IBBCEAS than the FTIR. So the kinetic method is not impacted by a loss of $NO_3$ due to photolysis. A precision has been added P.4 L.330: "**Finally, the intercomparison experiment shows that an eventual loss due to photolysis of $NO_3$ by the beam is not significant because it would have led to lower concentrations measured by the IBBCEAS than by the FTIR. In addition, for wavelength longer than 640 nm, which is the case here, Johnston et al., 1996 have shown that photolysis quantum yield is close to 0. It is thus expected that the photolysis of $NO_3$ in the used wavelength range is not occuring. Furthermore even though**

**NO₃ was slightly subject to photolysis locally, the volume enlightened by the probe beam is very small in comparison to the overall volume of the chamber (< 0.04%) and the homogenization system allowed a mixing with the rest of the volume.**"

680    The authors expect that wall losses of NO₃ are occurring. Nevertheless, as mentioned by the referee, the absolute kinetic method consists of measuring the decay of *trans*-2-butene and to use the NO₃ concentration measured during the decay. Then, NO₃ additional losses do not affect the rate constant determination. A sentence was added in the manuscript in order to precise this point (P.5 L.360): "**It is important to notice that the absolute kinetic method used consists of measuring the decay of *trans*-**

685    **2-butene for a known concentration of NO₃, and not the decay of NO₃ radicals for a known concentration of the VOC. The method is thus not affected by NO₃ additional loss processes (e.g. wall losses, reactions with NO₂ or with peroxy radicals). Only additional losses of the VOC would lead to an overestimation of the rate constant. This was checked prior to the experiments (i.e. before in the injection of N₂O₅) and no significant loss of the VOC was observed in the timescale of the**

690    **experiment (see below).**"

There is a very strong water vapor absorption in the wavelength range used. Nevertheless, all experiments have been performed in dry conditions and no water features were present on the spectra. A sentence has been had P.2.2 L.147:"**Absorption by water vapor may be very high under atmospheric conditions.**"

695    Aerosols can also affect the IBBCEAS technique at two levels: i) in general, deposition of aerosols on the mirrors lead to a decrease of their reflectivity. That has been dealt with the nitrogen purge in case of the generation of aerosols. ii) The absorption and scattering of the beam by aerosols can lead to a major decrease of the signal in the used wavelength range. In this study, no SOA was formed in the chamber, causing no such problem. In case of the use of the technique with a system which produces

700    SOA, experimental conditions have been modified by using the nitrogen purge close to the mirrors surface and by reducing the production of SOA (by reducing the concentration of precursor).

**5. It is not entirely clear how the effect of I0 stability has been determined. Is the 3% change in NO2 concentration seen at all concentrations, or is it a typical or worst case scenario? Can losses to walls**

705    **influence the analysis? The residual in Fig. 3 looks very much like the NO₂ absorption spectrum, so why does fitting with the NO₂ cross-section not eliminate these features better?**

The impact of $I_0$ stability has been evaluated by performing 2 types of experiments: First, the stability of the optical system has been verified. For this purpose, long term measurements of the signal have been done, leading to intensity variations lower than 0.3% and to very small baseline distortions. Then,

710    the impact of these baseline distortions on the quantification of the absorbing species has been evaluated by measuring the evolution in time of the concentration of a stable species, here NO₂. A mixture of 1 ppm of NO₂ in synthetic air was used. A sentence has been added in the manuscript (P.3.2 L.193): "**two types of experiments have been performed: first, the stability of the optical system has been verified. For this purpose, long term measurements of the signal have been performed, leading**

715    **to variations lower than 0.3% and to very small baseline variations. Second, to verify the impact of these variations on quantification of the absorbing species, experiments were conducted […] Then, a concentration of NO₂ was introduced into the chamber (mixing ratios ranging between 100 ppb and 1 ppm, depending of the experiment) and the signal I(λ) was measured**" and P.3.2 L.202: "**for a concentration of 1 ppm of NO₂.**"

720    It looks very unlikely that wall loss can influence the analysis, because the concentration is measured and decay would have been observed and $NO_2$ is very stable in the CSA chamber. In addition, $NO_2$ concentration is not constrained for the fit so in the case $NO_2$ concentration was not stable, the fitted concentration would be lower.

725    The referee is right when saying that the residual looks like $NO_2$ absorption spectra. It can be explained by a deformation of the baseline in time, due to the remoteness of $I_0$, which prevents the software to fit correctly the $NO_2$ spectra. It is likely that the residual is composed of both this baseline deformation and the small part of $NO_2$ cross section that is not correctly subtracted. The uncertainty generated by this phenomenon is said in the manuscript P.3.2 L.202 (up to 3 % for $NO_2$ concentration measurements).

730

**6. For the detection limit calculation, the authors should specify whether this calculation is for a purged or unpurged system.**

The detection limit was calculated without the purge system. It has been specified P.3.3 L.214 in the manuscript: "**and an unpurged system,**"

735

**7. The calculation of the detection limit is based on the signal-to-noise ratio for a single wavelength. However, the spectral fit is constrained by many independent signal measurements at different wavelengths. The authors should evaluate how the multiplex nature of the measurement affects the system's detection limit.**

740    We indeed used the classical method to determine the detection limit: it has been calculated on the signal-to-noise ratio for a range of wavelength between 655 nm and 666 nm, and we used the max. value of $NO_3$ absorption. This spectral range presents high absorption of $NO_3$ and weak absorption of $NO_2$, as said in the answer to the question 2, L.270.

It should be said that in real conditions, as shown in figure 4 in the manuscript, 6 ppt of $NO_3$ are very
745    close to the detection limit. The determination of the detection limit is thus coherent. Furthermore, this figure shows a spectrum with 630 ppb of $NO_2$, which is an important concentration. The detection limit is not degraded by the absorption of $NO_2$.

**8. Figure 6: At low pptv, it looks like there's still an IBBCEAS signal for $NO_3$ but no FTIR signal. Does**
750    **this divergence reflect a difference in instrument sensitivities, or equilibrium/heterogeneous chemistry, or some other cause? Please comment.**

By looking at the data, it seems that this case is for the first two points ($[NO_3]_{FTIR}$ = 3.0 ppt; $[NO_3]_{IBBCEAS}$ = 6.5 ppt and $[NO_3]_{FTIR}$ = 4.6 ppt; $[NO_3]_{IBBCEAS}$ = 6.7 ppt). For higher concentrations, the agreement is very good. It seems reasonable to think that the sensitivity of the FTIR measurement can explain this
755    difference: it is based on the measurement of $NO_2$ and $N_2O_5$. At this moment, $N_2O_5$ concentrations are small (approx. 8 ppb) which is close to the detection limit of the FTIR for this species (approx. 6 ppb). The uncertainty of this measurement is thus high, which can explain this difference at very small concentration of $NO_3$. Please note also that these concentrations are close to the detection limit of the IBBCEAS, so it will not be used in the frame of a kinetic experiment.

760

**9. L.313: Are spatial inhomogeneities in gas concentrations expected in the chamber?**

During experiments, a homogenization system is used to prevent from inhomogeneities of the mixtures. It is constituted in three parts: (i) an injection pipe, allowing injecting all along the chamber, ii) two fans, allowing a homogenization of gas inside the chamber and iii) a close-circuit homogenization pump, which samples in one extremity to inject the mixing back in the other one. Experiments have been conducted with an inert gas ($NO_2$) in order to measure the mixing time. Measurements were done in several points of the chamber. This system thus allows a mixing time inferior to a minute. The good agreement between the two measurement methods in the intercomparison experiment, which measure in two different areas, shows also that there are no major inhomogeneities.

A sentence has been had to explain it better in the manuscript P.2.1 L.61: "**It is equipped with a homogenization system which is made of i) an injection pipe (4 meters long, 1 cm diameter and regularly drilled with 1 mm holes) which allows to inject the reactants all along the chamber, ii) 2 stainless steel fans allowing a homogenization of gas inside the chamber and iii) a close-circuit Teflon pump connected at both ends allowing a recirculation of the gas mixing. This system allows a mixing time below one minute.**"

**10. Figure 7. $NO_3$ and butene losses are very rapid. Would data quality be increased by selecting reaction conditions to produce a slower reaction and more data points?**

For the kinetic of *trans*-2-butene, it was not necessary to increase the frequency of the measurement because the rate constant is relatively low. For faster kinetics, experimental conditions can be changed, in particular by reducing VOC concentration or increasing the measurement frequency.

**11. Most of the uncertainty in the kinetic data arises from the uncertainty in the FTIR results. Calibrated CIMS or PTR-MS might be better suited for co-measurement of VOCs. The authors may want to discuss these considerations for their kinetic studies.**

We fully agree with the referee that PTR-MS measurements might have been more precise. Unfortunately, this instrument was not available in our group when we performed the experiments.

**Minor corrections:**

**L.27: either "the NO3 radical has. . ." or "NO3 has. . ."**

It has been done.

**L.38: "one of the reasons for this. . ."**

It has been done.

**L.46: "progress has been made"**

It has been done.

**L.53: "For this purpose,"**

It has been done.

**L.63: "presented in detail"**

It has been done.

800 **L.76: "NO3 was"**

It has been done.

**L.101: "planar/concave"**

It has been done.

**L.106: "prevent adsorption of. . ."**

805 It has been done.

**L.115: Caption Caption should note that collimating lenses and curved mirrors are not shown on the beam injection side.**

It has been done.

**L.127: "LED current is fixed at 900 mA"**

810 It has been done.

**L.132: "to focus the beam"**

It has been done.

**L.143: hyphen in reference**

It has been done.

815 **L.175: "justifies...wide scale". It is unclear what is meant by this. That it should be measured over a wide wavelength range, or that it is not necessary to do so?**

The sentence has been changed to: "which justifies that it is necessary to measure it on a wide wavelengths range".

**L.182: "prior to each experiment"**

820 It has been done.

**L.196: "up to 3 % in NO2 concentration"**

It has been done.

**Figure 3. Change "residue" to "residual" in figure titles and legends.**

It has been done.

825 **L.280: "was first filled with dry synthetic air"**

It has been done.

**L.281: "Air Liquide NO2,"**

N20 (should be read N-20) is actually the name of the cylinder.

**L.287: NO2 and N2O5 concentrations**

830 It has been done.

**L.300: in IBI and in the spectral treatment**

It has been done.

**L389: performance**

It has been done.

835 **L.396: Clarify "from ppt to ppt range for NO3"**

It has been done.

**L.400: Monitoring NO3 radicals**

It has been done.

**L.409: intercomparison**

840 It has been done.

845

850

855

 **Answer to Anonymous Referee #2**

First of all, the authors would like to thank the anonymous referee for this discussion and its constructive comments, corrections and suggestions that ensued. We have carefully replied to all its comments and the paper has been improved following its recommendations. All technical corrections suggested by the referee have been carefully performed. Answers have also been provided for all
865 comments and changes have been performed accordingly. Please find below the answers to the comments:

**1. The kinetics analysis ignores any loss of NO3 other than bimolecular reaction of $NO_3$ with Alkene. This is most probably an over simplification. For example, what about wall loss, there could also be**
870 **diffusion outside of the analysis region. Using a simple model, the addition of a first order loss of $NO_3$, still results in a straight line from a plot of the change in alkene vs the product of [alkene].[NO₃] delta t (as in figure 8), however the retrieved rate coefficient is lowered and this impact gets worse as the wall loss increases. So if there is wall loss the retrieved rate coefficient does not equate to the bimolecular rate coefficient. How have the authors taken into account wall loss or any other first**
875 **order loss process? Have they modeled their system? Can they show what the quantitative impact on the retrieved rate coefficient will be? Have they made any attempt to evaluate first order losses of $NO_3$?**

We may have not explained clearly enough the kinetic method, because it appears that the referee did not understand it. $NO_3$ is indeed subject to losses other than the reaction with the VOC, for example
880 wall losses in the chamber, reaction with $NO_2$, with peroxy radicals… Nevertheless, this method consists of monitoring the decay of the *trans*-2-butene and not the one of $NO_3$, and this decay is plotted as a function of $[VOC] \times [NO_3] \times \Delta t$. Here, $[NO_3]$ is measured with the IBBCEAS technique as a function of the time (and not modelled), so we don't care about additional loss processes. What is important here is the measured concentration of $NO_3$ radicals by the time the consumption of the VOC is measured.
885 So, $NO_3$ can be consumed by other processes without affecting the result. Bu if the VOC is consumed by other loss processes (e.g. wall losses), this could affect the results. For this reason, *trans*-2-butene stability has been checked before every oxidation, and no significant loss beside dilution was observed. It allowed the authors making the hypothesis that *trans*-2-butene was only consumed by reaction with $NO_3$.

890

**2. The authors claim that there is good agreement with literature, however, their rate coefficient is the fastest that has been reported in the literature. Also, given that this rate coefficient is probably not simply the rate coefficient for the bimolecular process, most probably a lower limit as result of other first order loss proves, the true bimolecular rate coefficient in their system is likely to be even**
895 **faster. Faster rate coefficients normally worry kineticists, if one looks at the absolute rates reported in table 3 the unweighted average is $3.87 \times 10^{-11}$ with a standard deviation of $0.12 \times 10^{-11}$ i.e. statistically the reported rate coefficient is higher than the current experimental database. Can the authors explain why this is the case?**

We are a bit surprised by this comment. Isn't $(3.87 \pm 0.12)$ in agreement with $(4.13 \pm 0.78)$? Agreement
900 between values has to be considered by taking into account the uncertainties, otherwise it has no sense. And by taking them into account, our determination is in agreement with the unweighted average of absolute determinations and with the IUPAC recommended value. But it is also in agreement with all individual absolute determinations. The only disagreement which has been noticed

is with relative determinations which are significantly lower than other determinations and where not taken into account for the IUPAC recommended value. In conclusion the authors affirm that our determination is in good agreement with literature.

**3. In Table 1, Deff is given as 82 which is simple the geometric distance, if I am not mistaken. Will that be the case? A purge flow is used to protect the mirrors and this will impact the effective path length, this is also likely to be a function of pressure and flow of purge gas. Have experiments been performed to quantify the Deff as a function of purge gas flow rate and pressure?**

All experiments presented here were performed without a purge flow so the effective distance was considered to be the geometric distance (82 cm). A sentence clarifying this statement has been added in the manuscript (P.2.2 L.111): "**Nitrogen flush was not used in this study, but is available for further type of experiments.** "

Nevertheless, experiments have been conducted to quantify the impact of purge gas flow and pressure:

- Due to nitrogen purge, pressure is only slightly rising: the purge flow is 300 mL min$^{-1}$, and 300 mL represents a variation of 0.03 % of the total volume of the chamber. This flow is compensated by the sampling of measurement instruments. No pressure variation was ever measured due to the purge gas flow. Nevertheless, characterization experiments using a known $NO_2$ concentration were conducted, in order to quantify the impact of major pressure variation on the pathlength. They showed no significant impacts of pressure on the pathlentgth. Pressure variations of 10 mbar in the chamber lead indeed to decrease of the path length of less than 1 %.
- Using also a known concentration of $NO_2$, experiments have been performed by adding different flows of purge, in order to determine an efficient flow, i.e. which does not induces a major decrease of the pathlength and a progressive dilution in the measurement area. They showed that the optimal flow is 300 mL min$^{-1}$, because it efficiently protects the mirrors and induce no long term dilution. With this flow, Deff is decreasing to 62 cm (as it is written in the manuscript L.110), adding an uncertainty of 5 % to the nitrate radical concentration. A sentence to explain these experiments has been added in the manuscript P.2.2 L.108: "**This flow rate has been optimized in order to efficiently protect the mirrors while limiting the dilution of the mixture in the measurement area.**"

**4. On line 290 the authors state "these parameters were therefore precisely monitored during the experiment leading to the value of 2.17 × 10-11 cm3.molecule-1 at 298K and at 1030 mbar." Is that the value of the equilibrium constant? If so, what are the errors? They need to be stated. Can the authors show that within experimental error that agrees with those reported in Atkinson et al., 2004, indeed how does it compare to the recommended IUPAC /JPL recommended values?**

The value of 2.17 × 10$^{-11}$ cm$^3$.molecule$^{-1}$ is indeed the equilibrium constant. This value was not measured in our study, but calculated using the parametrization recommended by the IUPAC (Atkinson et al., 2004) at a pressure and temperature which were measured during the experiments. So, here, it appears that there is a misunderstanding. By "parameters", we mean temperature and pressure and not the rate constant. In order to make it clearer, the sentence has been modified P.4 L. 305 in the manuscript: "**These two parameters were therefore precisely monitored during the experiment**

**allowing calculating an equilibrium constant of 2.17 × 10$^{-11}$ cm$^3$.molecule$^{-1}$ at 298K and at 1030 mbar, using IUPAC database parameters (Atkinson et al., 2004).**"

**5. In Figure 6, the authors report linear regression between the FTIR and the BBCEAS. Can they provide errors on the slopes? Also include a description of those errors, e.g. are they 1 sigma just from the linear fit, or do they take into other experimental errors?**

Errors on the slopes are now provided. The values were calculated to be 1.0 ± 0.2 for NO$_2$ and 1.1 ± 0.3 for NO$_3$ by considering the statistical error on the slope, which is twice the standard deviation, and the sum of the systematic relative errors on FTIR and IBBCEAS measurements.

To explain it, sentences have been written in P.4 L.313: "**Here, the overall uncertainty was calculated as the sum of the statistical error on the slope (twice the standard deviation, 4 %) and systematic errors on FTIR (i.e. on IBI$_{NO2}$, 4 %) and IBBCEAS measurements (which includes uncertainties on NO$_2$ cross sections and on the mirrors reflectivity, 9%).**"and in P.4 L.320 "**The error is calculated with the same method as for NO$_2$.**"

**6. Finally what are the absolute errors on the rate coefficient? The authors report a simple error analysis based on the line of best fit, i.e. what is the total error? The authors need to take into account errors in flows, absorption cross section etc etc**

The authors have considered that the statistical error, calculated as twice the standard deviation on the linear regression and which takes into account the dispersion of the experimental points, is mainly due to spectra treatment uncertainties. However, we agree that this statistical error does not include systematic ones, such as errors on cross sections. So we have added the uncertainty on NO$_3$ concentration. As described in the manuscript, this uncertainty was estimated to be 9% and corresponds to the sum of NO$_2$ and NO$_3$ cross sections errors and the uncertainty of NO$_2$ concentration for reflectivity measurement. By summing the statistical error (10%) and the uncertainty on NO$_3$ concentration (9%), we obtain an overall uncertainty of 0.78 × 10$^{-13}$ cm$^3$ molecule$^{-1}$ s$^{-1}$. This has been corrected in the manuscript L.391 and in Table 3. It must be noticed that because *trans*-2-butene concentration is both on x and y axes, the error on IBI is not considered here.

An explanation has been added P.5 L.392: "**The uncertainty on the rate constant was estimated as the sum of the relative uncertainties on NO$_3$ concentrations and twice the standard deviation on the linear regression.**"